# Bridging Domain Gaps in Computational Pathology: A Comparative Study of Adaptation Strategies

**DOI:** 10.3390/s25092856

**Published:** 2025-04-30

**Authors:** João D. Nunes, Diana Montezuma, Domingos Oliveira, Tania Pereira, Inti Zlobec, Isabel Macedo Pinto, Jaime S. Cardoso

**Affiliations:** 1Institute for Systems and Computer Engineering, Technology and Science, 4200-465 Porto, Portugal; tania.pereira@inesctec.pt (T.P.); jaime.cardoso@inesctec.pt (J.S.C.); 2Faculty of Engineering, University of Porto, 4200-465 Porto, Portugal; 3IMP Diagnostics, 4150-146 Porto, Portugal; diana.felizardo@impdiagnostics.com (D.M.); domingos.oliveira@impdiagnostics.com (D.O.); isabel.macedo.pinto@impdiagnostics.com (I.M.P.); 4Cancer Biology and Epigenetics Group, Research Center of Portuguese Oncology Institute of Porto/RISE@Research Center of Portuguese Oncology Institute of Porto (Health Research Network), Portuguese Oncology Institute of Porto/Porto Comprehensive Cancer Centre Raquel Seruca, R. Dr. António Bernardino de Almeida, 4200-072 Porto, Portugal; 5FCTUC—Faculty of Sciences and Technology, University of Coimbra, 3004-516 Coimbra, Portugal; 6Institute of Tissue Medicine and Pathology, University of Bern, 3008 Bern, Switzerland; inti.zlobec@unibe.ch

**Keywords:** domain adaptation, weakly supervised learning, consistency regularization, multiple instance learning, computational pathology

## Abstract

Due to the high variability in Hematoxylin and Eosin (H&E)-stained Whole Slide Images (WSIs), hidden stratification, and batch effects, generalizing beyond the training distribution is one of the main challenges in Deep Learning (DL) for Computational Pathology (CPath). But although DL depends on large volumes of diverse and annotated data, it is common to have a significant number of annotated samples from one or multiple source distributions, and another partially annotated or unlabeled dataset representing a target distribution for which we want to generalize, the so-called Domain Adaptation (DA). In this work, we focus on the task of generalizing from a single source distribution to a target domain. As it is still not clear which domain adaptation strategy is best suited for CPath, we evaluate three different DA strategies, namely FixMatch, CycleGAN, and a self-supervised feature extractor, and show that DA is still a challenge in CPath.

## 1. Introduction

Computational Pathology (CPath) is one of the critical fields where Machine Learning (ML) is revolutionizing clinical practice by increasing productivity, reducing costs, improving the precision of diagnosis, and enabling more effective treatments. But a challenge with microscopy imaging, from which Hematoxylin and Eosin (H&E)-stained Whole Slide Images (WSIs) is one of the most widely available modalities, is the high variability in image chromaticity and morphological appearance, aside from batch effects and hidden stratification that prevent generalization beyond the training data distribution. Deep Learning (DL) is the go-to solution for WSI analysis, with the top-down approach one of the most common methods [1]. The strategy consists of splitting the gigapixel WSIs into multiple lower-dimensional tiles, subsequently encoded into tile-level features (or scores), followed by an aggregation function for slide-level predictions. WSI analysis is thus naturally a Multiple Instance Learning (MIL) paradigm, and its success strongly depends on the quality of tile-level representations. Indeed, although ImageNet pre-training and self-supervised learning on unlabeled histopathology datasets were shown to be successful for tile-level feature extraction [2,3,4,5,6], task-specific fine-tuning could translate to better results [1,7,8,9].

DL relies on large volumes of diverse and annotated datasets that accurately reflect the statistical distribution of the entire population. However, these datasets are often difficult to collect and annotate in CPath, leading to biased solutions that often fail even for small distribution shifts. Therefore, algorithms with outstanding results on independent and identically distributed (IID) samples frequently depend on dataset-specific attributes, i.e., spurious correlations that do not hold in related but distinct distributions (i.e., domains). Nevertheless, a common setting is to have access to a significant number of annotated samples from one or multiple source distributions and another (frequently smaller) dataset representing a target distribution with little to no annotations for which we want to generalize, the so-called Domain Adaptation (DA) paradigm. Indeed, factors such as stain variability, blurriness, and resolution inconsistencies severely affect the generalization of DL models for CPath. Nonetheless, several strategies have been proposed to address these issues, like Generative Adversarial Networks (GANs) to deal with stain variability or adversarial learning, which learn invariant features. In this work, we focus on the challenging scenario of single-source DA, which aims to transfer knowledge from a single-source distribution to a target domain. However, as it is still not clear which DA strategy is best suited for CPath, we evaluate and compare three different approaches, namely a weakly supervised learning (WSL) strategy based on consistency regularization (FixMatch [10]), a domain transfer approach with CycleGAN [11], and a method based on self-supervised feature extraction with a foundation model (Phikon [12]). We explore the complementarity of these methods and compare these strategies in semi-supervised DA (SSDA) and unsupervised DA (UDA) to clarify how performance degrades when no annotations from the target distribution are available. We thus expect to shed light on the effectiveness and limitations of each method in addressing DA in CPath. Moreover, to the best of our knowledge, we are the first to address the curse of domain-invariant representations in CPath. Besides this introduction, this work is structured as follows: In Section 2, we start by a review of works similar to ours; then, in Section 3, we discuss the methods we consider in this work, while in Section 4 we describe the experimental details of this work. After, in Section 5, we present the results and provide a discussion in Section 6, and, finally, in Section 7, we unveil the main conclusions of this work.

## 2. Related Work

The task of DA usually assumes that the conditional distribution of the labels given the data remains stable across source and target domains ($P\left(Y|X_{target}\right)=P\left(Y|X_{source}\right)$) but that the marginal distributions can change ($P\left(X_{target}\right)\neq P\left(X_{source}\right)$). This is the so-called covariate shift assumption and it motivates DA techniques from different angles. For example, reweighting samples from the source domain to match the target distribution [13], learning representations that are transportable across domains [14], using optimal transport to map the source distribution to the target domain [15], or the widely adopted strategy of learning domain-invariant representations [16]. However, a limitation is that the covariate shift assumption could be violated, meaning domain invariant representations hinder the generalization of the model to the target domain [17]. This is referred to as the curse of domain-invariant representations, with recent approaches empirically demonstrating the effectiveness of consistency regularization [18,19] to tackle this issue.

Specifically, in CPath, Ren et al. [20] suggest domain-adversarial networks for a prostate classification task. Koohbanani et al. [21] propose a multi-task learning framework for oral squamous cell carcinoma and breast cancer metastasis classification that combines SL with self-supervised pretext tasks to learn invariant representations without requiring annotations from the target domain. In turn, Abbet et al. [22] design a self-supervised method with curriculum learning to learn invariant features for colorectal cancer tissue detection, whereas Marini et al. [23] suggest a robust stain color augmentation technique, while also demonstrating its generalization effectiveness when combined with domain adversarial neural networks. Other approaches focus on Generative Adversarial Networks (GANs) for stain transfer between domains [24,25]. Yet, the usefulness of the various DA techniques available in computer vision still requires clarification in CPath. Moreover, to the best of our knowledge, no previous method in CPath addresses the curse of domain-invariant representations.

## 3. Materials and Methods

In this section, we present the methods that we address in this work. We start by detailing the theoretical foundation of the method based on FixMatch, followed by the formal description of CycleGAN, and then the self-supervised feature extraction strategy. We opt for these methods, as they can work without multiple source domains. We also describe the aggregation function we consider to evaluate the learned features. Figure 1 illustrates the methods we consider in this work.

### 3.1. Domain Adaptation with FixMatch

To learn effective tile-level features, we resort to two different approaches: (a) SL using only source domain data annotated at the tile level, and (b) a consistency regularization approach, namely FixMatch [10], using both source and target domains. Consistency regularization is based on the smoothness assumption of semi-supervised learning (SmSL), stating that close data points in the input space are likely to share the same label. Therefore, considering a label-preserving transformation, *t*, over some transformation space, T, the estimated labels for a given sample, *x*, and its transformed view, t(x), should be the same. In the context of SSDA, strong data augmentations could be interpreted as creating novel domains [18]. Therefore, consistency regularization promotes features that generalize well across domains [18]. Moreover, some augmented samples may fall in higher-density regions over the distribution of the source domain(s), meaning empirical risk minimization on the source distribution with consistency regularization on samples from the target domain should also minimize the error on the target distribution [18]. Fixmatch [10] enforces consistency by minimizing the cross-entropy between the most confident predictions for weakly augmented views of an unlabeled input and a strongly augmented version of the same sample. In essence, the strategy suggests a two-term compound loss, one for annotated data and one for unlabeled inputs.

Given two sets of labeled, $L=\{\left(x_{l}^{1},y_{l}^{1}\right),\ldots ,\left(x_{l}^{N},y_{l}^{N}\right)\}$ (source domain), and unlabeled, $U=\{\left(x_{u}^{1},y_{u}^{1}\right),\ldots ,\left(x_{u}^{M},y_{u}^{M}\right)\}$ (out of distribution target domain), images, sampled, respectively, in batches of size *B* and μB, with μ the proportion of unlabeled samples, FixMatch starts by computing a pseudo-label for unlabeled images. This is achieved by computing hard pseudo-labels, ${\hat{q}}_{u}^{m}=\arg\max\left(q_{u}^{m}\right)$ from the conditional distribution of the targets given a weakly augmented view of the unlabeled image, $t(x_{u}^{m})$, (or unlabeled pixels in the case of partially annotated images), $q_{u}^{m}=p\left(y_{u}^{m}|t\left(x_{u}^{m}\right)\right)$. Then, consistency is imposed by minimizing the cross-entropy between the conditional distribution of the class given a strongly augmented view of the input, $q_{u}^{\prime m}=p\left(y_{u}^{m}|t^{\prime }\left(x_{u}^{m}\right)\right)$, and confident pseudo-labels, $\max(q_{u}^{m})>\tau$:(1)$$L_{u}=\frac{1}{\mu B}\sum_{m=1}^{\mu B}=1\left(\max\left(q_{u}^{m}\right)>\tau \right)H\left({\hat{q}}_{u}^{m}|q_{u}^{\prime m}\right),$$
with ${\hat{q}}_{u}^{m}=\arg\;\max\left(p\left(y_{u}^{m}|t\left(x_{u}^{m}\right)\right)\right)$, τ a confidence threshold, H the cross-entropy loss, and 1 the indicator function.

For labeled images, the standard cross-entropy loss is adopted:$$L_{s}=\frac{1}{B}S\sum_{b=1}^{B}H\left(p_{b}|p\left(y|t\left(x_{l}^{b}\right)\right)\right)$$
with $p_{b}$ the ground truth (one-hot encoded) label.

The FixMatch objective thus becomes:(2)$$L=L_{s}+\lambda L_{u}$$
where λ is a hyperparameter weighting the relative contribution of the unsupervised loss.

An advantage of consistency regularizatoin with FixMatch is that it is effective in dealing with the curse of domain-invariant representations [18,19] where the assumption that the conditional distribution of the labels given the data remains stable across source and target domains ($P\left(Y|X_{target}\right)=P\left(Y|X_{source}\right)$) is violated.

### 3.2. Domain Transformation Using CycleGAN

CycleGANs enable domain translation with unpaired images by use of cycle-consistency. The task of transferring images from source, *X*, to target domain, *Z*, involves a generator, *G*, and a discriminator, $D_{1}$, to learn a mapping, G:X→Z, where *G* tries to generate images G(x) that resemble images from domain *Z* [11]. In turn, the role of the discriminator is to distinguish between real, *z*, and generated samples, G(x). The networks are optimized adversarially with the following objective (*G* tries to minimize the objective while $D_{1}$ tries to maximize it):(3)$$L_{GAN}\left(G,D_{1},x,z\right)=\log D_{1}+1-\log D_{1}\left(G\left(x\right)\right)$$

In theory, adversarial training can produce outputs with the same distribution as the target domain. However, provided the discriminator objective is fulfilled, the generator can transform the input image into any image from the target domain [24]. When dealing with unpaired image-to-image translation, cycle-consistency ensures that, when an image is translated from one domain to another, it can later be translated back to its original domain, maintaining its essential features and structure [11]. This is achieved through a second generator, *F*, and discriminator pair and enforcing F(G(x))≈x. The cycle-consistency loss is defined as(4)$$L_{cycle}=E_{x}[||G\left(F\left(x\right)\right)-{x||}_{1}]+E_{z}[||F\left(G\left(z\right)\right)-{z||}_{1}]$$

### 3.3. Self-Supervised Feature Extraction

Another promising and growing direction to extract tile-level features is domain-specific foundation models pre-trained with self-supervised learning on large-scale histopathology datasets [6,12]. These models were shown to be significantly more successful than ImageNet pre-trained models [5] and can be directly used for feature extraction or fine-tuned. In this work, we explore Phikon [12], a Vision Transformer (ViT) feature extractor, pre-trained on 40 million H&E-stained tiles from TCGA [26,27] using Masked Image Modeling (MIM) and the iBOT framework [28]. We then compare the generalization of this strategy with FixMatch [10], CycleGAN, and the supervised baseline.

### 3.4. Evaluation Protocol: Grading of Colorectal Biopsies and Polypectomies

We focus on the task of grading colorectal biopsies and polypectomies (excluding surgical specimens) into three classes: non-neoplastic (NNeo), low-grade (LG) lesions, and high-grade (HG) lesions (high-grade dysplasia and adenocarcinoma) [9]. To compare the effectiveness of the feature learning approaches, we consider training a slide-level classifier using a state-of-the-art aggregation function, namely TransMIL [2]. We evaluate the representations in SSDA and UDA. In the semi-supervised setting, we assume access to a small amount of slide-level annotated data from the target domain used to optimize a slide-level classifier, whereas, in the unsupervised setting, we have only labels from the source domain for training (at both tile and slide-level). Moreover, we consider an additional paradigm wherein we train TransMIL using a combination of samples from both domains.

## 4. Experimental Details

Code to reproduce the experiments is available at the following URL: https://github.com/joao-nunes/domain-adaptation-cpath (accessed on 28 April 2025).

### 4.1. Datasets

We resort to two H&E-stained WSI datasets, namely CRS10K (IMP Diagnostics, Porto, Portugal) [9,29,30] (available online: https://doi.org/10.25747/fb1q-j507 (accessed on 28 April 2025) and BernCRC (IGMP, Bern, Switzerland) containing samples of colorectal biopsies and polypectomies (excluding surgical specimens) with annotations for the dysplasia grade into Non-neoplastic (NNeo); Low-grade (LG); and High-grade (HG) lesions. The slides from CRS10K were digitized with Leica GT450 WSI scanners, at 40× magnification. All samples from CRS10K are annotated at the slide level, while roughly 9% of the WSIs (967) also contain instance-level annotations. From the annotated tiles of CRS10K, we reserve 200 WSIs for testing and 200 WSIs for validation, while the remaining data are used to train a tile-level classifier for subsequent feature extraction. In turn, BernCRC was digitized with 3DHistech P250 WSI scanners and is only labeled at the slide level. Figure 2 presents sample H&E-stained tiles from each dataset. The staincolor distribution shift between source and target data is evident.

### 4.2. Data Pre-Processing

As in the work by Neto et al. [9], to isolate the colorful tissue from the meaningless white background, the pre-processing of the slides consists of an automatic tissue segmentation with Otsu’s thresholding on the saturation (S) channel of HSV color space, leading to the separation between tissue regions and the background. Subsequently, a 32×32 down-sampled version of the generated mask is used to extract WSI tiles with dimension 512×512 pixels at 40× magnification and containing only tissue (i.e., a 100% tissue threshold is used).

### 4.3. Data Augmentation

For both the supervised baseline and FixMatch strategy (strong augmentations), we adopt a label-preserving data augmentation policy that was previously shown to be successful in CPath [8,31]. Table 1 presents the details of the transformations we use and the corresponding hyperparameters. For data augmentation, we use the *python* libraries *Albumentations* version 1.3.0, and *scikit-image* version 0.19.3. Figure 3 shows some sample H&E-stained tiles from CRS10K before and after data augmentation.

### 4.4. Training

#### 4.4.1. Tile-Level Feature Learning

We make use of ResNet-34 as the feature extractor, pretrained on ImageNet and fine-tuned using both SL on the source domain and FixMatch.

As first suggested in the work by Oliveira et al. [29], we treat the grading of colorectal biopsies and polypectomies as an ordinal regression problem and adopt a soft version of QWK [32] as the loss function, which weighs misclassifications differently depending on the distance between the predicted and ground truth label.

As in the work by Neto et al. [9], we adopt as an optimizer stochastic gradient descent (SGD) with momentum =0.9, a learning rate of $1\times 10^{-4}$, and a weight decay of $3\times 10^{-4}$. To train the FixMatch method, we consider λ=0.3 and τ=0.9. We use a batch size, *B*, of 32 for the annotated samples and μ=0.5, i.e., μB=16, for the unlabeled inputs. In addition, we include a warm-up period of 5000 iterations before introducing the FixMatch regularization term. To generate weakly augmented views, we only normalize the input with mean =0.5, and std =0.1. We adopt stratified k-fold cross-validation (k = 5) for BernCRC data and divide the data into train, validation, and test splits using, respectively, 67.5%, 12.5%, and 20% of the data. The tile-level feature learning experiments are implemented in *python* 3.10.8 with *Pytorch* version 2.0.0 and a single Tesla V100 32GB GPU. Due to computational limits and long run-times, hyperparameter optimization was limited. Our hyperparameters thus derive from similar works and from practicalities, and are likely to be sub-optimal.

#### 4.4.2. CycleGAN

We use the architectures suggested in the work by Zhu et al. [11]. The generator consists of a stride-1 convolution with kernel size equal to 7, followed by two convolutions with stride-2 and kernel size 3, 15 residual blocks, followed by two convolutions with up-sampling, and a final convolution with stride 1 and kernel size 7. All convolutions are followed by instance normalization and ReLU activation. For the discriminator network, we are inspired by PatchGAN [24]. As in de Bel et al. [24], the discriminator consists of four convolutional layers with 64, 128, 256, and 512 filters, respectively, and with kernel size of 4 and stride 2. A final convolution ensures the output is a single filter map. All convolutions except for the last were followed by an instance normalization layer and a leaky ReLU with the leak parameter equal to 0.2. As data augmentation, we consider only normalization with mean = 0.5 and std = 0.5.

#### 4.4.3. TransMIL

We use the official code base when implementing TransMIL [2]. We train the model for 50 epochs using LookAhead [33] optimizer with RAdam [34], a batch size *B* equal to 1, learning rate $1\times 10^{-4}$, and weight decay $1\times 10^{-5}$. For testing, we select the model with the best validation accuracy. For the slide-level aggregation experiments, we select a subset of 1000 WSIs (CRS1K) from the CRS10K training data (source domain) annotated at the slide level. We perform validation and testing using the publicly available validation and test sets from CRS10K. In turn, regarding the target domain (BernCRC data), we adopt a stratified k-fold cross-validation (k = 5) and divide the data into train, validation, and test splits using, respectively, 67.5%, 12.5%, and 20% of the data. To optimize TransMIL using samples from both domains, we train the model using two batches of data at each iteration (one from each domain). For the loss term corresponding to each mini-batch, we perform hyperparameter tuning with a random search using the CRS1K training set and one of the five folds from BernCRC and select the values with the highest validation accuracy. We observed $\theta_{1}=0.3$ (CRS1K) and $\theta_{2}=0.7$ (BernCRC) translated to the best validation results. The experiments for slide-level aggregation are implemented in *python* 3.10.8 with *Pytorch* version 2.0.0 and a single Nvidia GeForce 1080ti 11 GB GPU.

### 4.5. Evaluation Metrics

#### 4.5.1. Quadratic Weighted Kappa

The Quadratic Weighted Kappa (QWK) measures the agreement between two raters. This metric typically varies from 0 (random agreement between raters) to 1 (complete agreement between raters). It is an appropriate metric for ordinal data as it weighs misclassifications differently depending on the distance between the predicted and actual classes, according to the following equation:(5)$$QWK=1-\frac{\sum_{i,j=1}^{n}w_{i,j}x_{i,j}}{\sum_{i,j=1}^{n}w_{i,j}m_{i,j}}$$
where $w_{i,j}$ belongs to the weight matrix, $x_{i,j}$ belongs to the observed matrix, and $m_{i,j}$ are elements in the expected matrices. The n×n matrix of weights *w* is computed based on the difference between the actual and predicted class, as follows:(6)$$w_{i,j}=\frac{{\left(i-j\right)}^{2}}{{\left(n-1\right)}^{2}}$$

#### 4.5.2. Accuracy

Accuracy is a metric that measures how often an ML model is correct. It is defined as the proportion of correct predictions over the total number of predictions:(7)$$Acc=\frac{TP+TN}{TP+FP+TN+FN}$$

With TP the number of true positive predictions, TN the true negative predictions, FP the number of false positives, and FN the number of false negative predictions.

#### 4.5.3. Sensitivity

Sensitivity or True Positive Rate (TPR) defines the number of true positives that are correctly identified by the model:(8)$$Sens=\frac{TP}{TP+FN}$$

#### 4.5.4. Specificity

Specificity or True Negative Rate (TNR) measures the proportion of true negatives that are correctly identified by the model:(9)$$Spec=\frac{TN}{TN+FP}$$

#### 4.5.5. Precision

Precision quantifies the proportion of true positive predictions among all positive predictions made by the model:(10)$$Prec=\frac{TP}{TP+FP}$$

#### 4.5.6. F1 Score

The F1 score is the harmonic mean between precision and recall:(11)$$F1=2\times \frac{Precision\times Recall}{Precision+Recall}$$

#### 4.5.7. Area Under the Receiver Operating Characteristic Curve (AUC)

The Area Under the Receiver Operating Characteristic Curve (AUC) summarizes the trade-off between the TPR and false positive rate (FPR) across all possible decision thresholds. An ML model with perfect discriminative ability will have an AUC of 1, while a model without discriminative power will have an AUC of 0.5. AUC presents two important advantages: it is scale-invariant as well as classification threshold-invariant.

## 5. Results

### 5.1. Source Domain Results

Table 2 and Table 3 report the results of model evaluation on IID samples. We observe high performance regardless of the feature extractor learning strategy and that incorporating unlabeled samples from an additional data distribution leads to slight improvements over all metrics, thus hinting that the adopted consistency regularization technique enables learning more generalizable representations.

### 5.2. Unsupervised Domain Adaptation

Table 4 presents the Fréchet Inception Distance (FID) between BernCRC (target) transformed to CRS10K (source) and BernCRC (target) to CRS10K (source). As we observe, the FID metric is smaller between the source and transformed target than between the source and target, thus suggesting the visual quality and realism of the domain transformation approach.

Table 5 and Table 6 report the results in the UDA paradigm for, respectively, colorectal tissue classification into NNeo, LG, HG, and binary classification. Figure 4 summarizes the main metrics (QWK, Accuracy, and Binary Accuracy) for the UDA task. Single Source DA is still a challenging scenario as, although we observe outstanding results when the training and testing data originate from the same distribution, the performance greatly decreases in the target domain.

### 5.3. Semi-Supervised Domain Adaptation

Table 7 and Table 8 report the results in the SSDA paradigm for, respectively, grading of colorectal biopsies and polypectomies (NNeo, LG, and HG) and binary classification. Figure 5 summarizes the main metrics (QWK, Accuracy, and Binary Accuracy) for the SSDA task.

## 6. Discussion

### 6.1. Unsupervised Domain Adaptation

The results suggest that FixMatch is a better strategy to learn more generalizable tile-level representations, as in the UDA paradigm, the MIL aggregator (TransMIL) achieves higher performance (e.g., an improvement of +14.09% in the QWK metric, +12.78% in accuracy, and +10.13% in binary accuracy for FixMatch) when compared with only SL on the source domain. We argue that this is due to the ability of FixMatch to learn representations that are transferable across the two domains. Nonetheless, the best results are observed with CycleGAN (e.g., an improvement of 16.46% in QWK, 20.63% in accuracy and 12.18% in binary accuracy when compared with supervised learning using only source domain data) showing that transforming images from one domain to another is a competitive strategy. Nevertheless, there is still potential for improvement. We suggest that the limitations of Cycle-GAN derive from synthetic image generation artifacts and from tissue-specific artifacts that are ineffectively modeled, like blurriness, tissue folds, or air bubbles. Regarding the FixMatch strategy, although it enables learning more general features shared across the two domains, it still cannot completely model the stain variability, as it focuses on data augmentation instead of modeling the domain-specific stain variability. This might suggest that these methods are complementary, as while the FixMatch strategy enables learning features shared across the two domains, Cycle-GAN mitigates stain variability. Another finding of this work is that data augmentation significantly improves performance compared to no data augmentation, with a 32.78% increase in QWK and a 21.1% increase in accuracy. However, a limitation of this study is that we only tested one combination of hyperparameters for data augmentation due to the long run times, which may lead to suboptimal results. Furthermore, it would be beneficial to compare our results with the existing literature. Unfortunately, there is limited research on domain adaptation in computational pathology, which makes direct comparisons difficult.

### 6.2. Semi-Supervised Domain Adaptation

The best results are achieved when slide-level-labeled samples from the target domain are available to optimize the slide-level classifier. In this setting, we achieve competitive performance despite the strategy used to learn tile-level representations. Nonetheless, the MIL aggregator with the FixMatch feature extractor achieves superior generalization (e.g., QWK increases +1.95%, and accuracy improves +0.95% (BernCRC)), while a top accuracy is also achieved when training TransMIL with source (CRS1K) and target (BernCRC) data. Compared with the UDA scenario, we observe significant improvements in performance in all metrics, which reflects that, despite the same tile feature extractor, the MIL paradigm is still subject to the simplicity bias, meaning the model is prone to depend on domain-specific spurious correlations. In turn, CycleGAN achieves a QWK of 0.6088 and an accuracy of 83.24 %.

Another finding of this work is that task-specific fine-tuning should be the preferred strategy to learn effective tile-level representations for grading of colorectal biopsies and polypectomies as the fine-tuned models compare favourably with self-supervised learning on large-scale histopathology datasets. However, the performance of the optimized WSI classifiers on top of Phikon tile embeddings is quite competitive with the fine-tuned models, hinting that further performance improvements can be observed after fine-tuning Phikon.

Overall, DA in CPath is a challenging objective, and the problem should be tackled from two fronts: On the one hand, there is the need to learn effective tile-level representations; on the other hand, the aggregation of instance features into a robust global slide representation is still prone to the simplicity bias. Therefore, DA and generalization techniques should also be adopted at this level.

## 7. Conclusions

In this work, we show that a WSL method based on consistency regularization, namely FixMatch, is a promising strategy for UDA in CPath, as it achieves superior generalization when compared to SL when no labeled data from the target domain are available. However, we observe the best results in UDA with CycleGAN, hinting that transforming images from source to target domain enables the reduction in stain variability that hinders DL model performance. In turn, we find that slide-level labels from the target domain are essential to learning an optimal slide-level classifier in the target domain, thus demonstrating that, despite the tile-level feature encoder, slide-level classifiers are still subject to the simplicity bias. Overall, generalizing to out-of-distribution samples is still challenging in CPath, and despite the promises of DA, there are still a few limitations, like hidden stratification and batch effects, stain, blur, and digitization artifacts, or over-representation of certain subgroups in the training set. Future work should focus on comparing other DA strategies and better understanding these issues to develop more robust models for WSI analysis. Another possible direction is the combination of the FixMatch strategy with CycleGAN, taking advantage of the complementarity of the techniques for more robust classifiers. Other possible future directions are multitask and multimodal learning. Whereas multitask learning allows for learning more general features through a shared backbone in multitask models [35,36], multimodal learning allows the model to take advantage of the complementarity of multiple data modalities like molecular profiling and H&E-stained WSI data [37]. However, this comes with the challenge of collecting paired multimodal data.

## Figures and Tables

**Figure 1 sensors-25-02856-f001:**
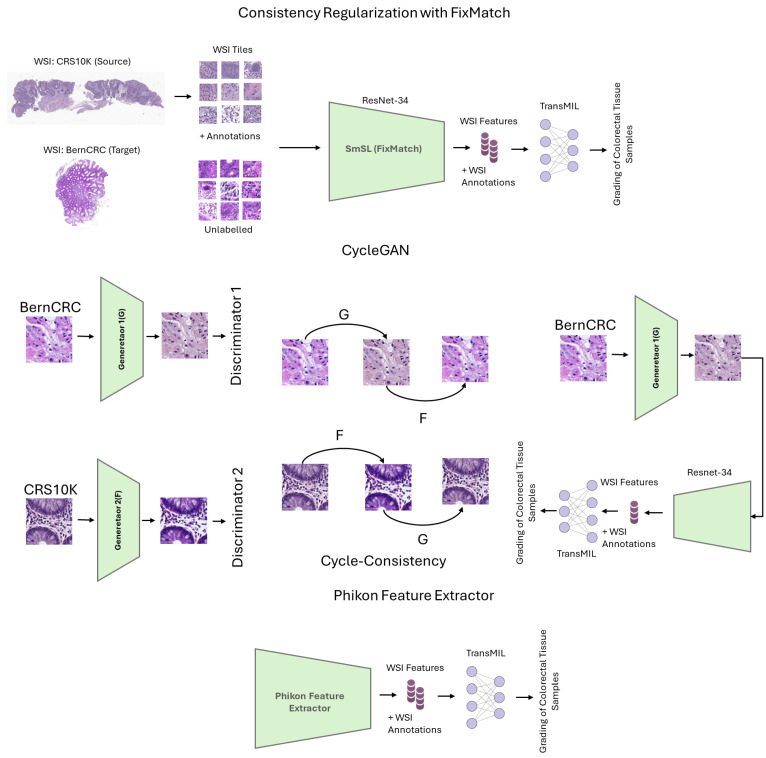
Illustration of the proposed framework for weakly supervised domain adaptive colorectal tissue classification. Top: Fixmatch based strategy; Middle: CycleGAN; Bottom: Phikon feature extractor.

**Figure 2 sensors-25-02856-f002:**
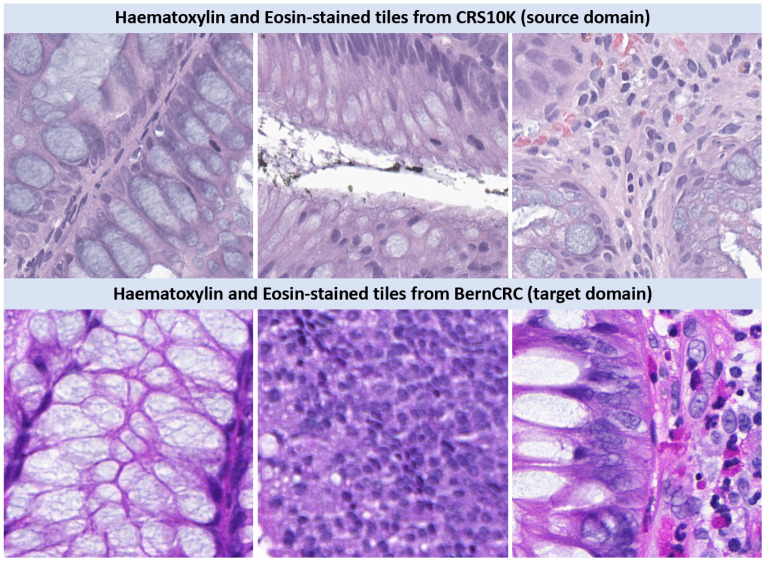
H&E-stained images from CRS10K—IMP Diagnostics (source) and BernCRC (target). The stain color distribution shift between source and target data is quite evident.

**Figure 3 sensors-25-02856-f003:**
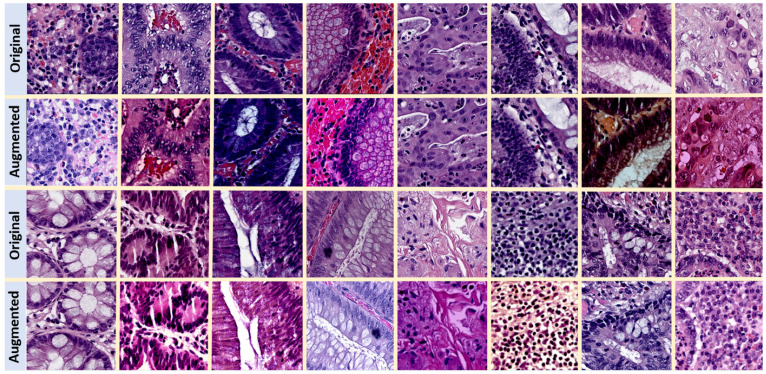
Illustration of data augmentation on H&E-stained images from CRS10K. Stain color variation after data augmentation is the most salient feature.

**Figure 4 sensors-25-02856-f004:**
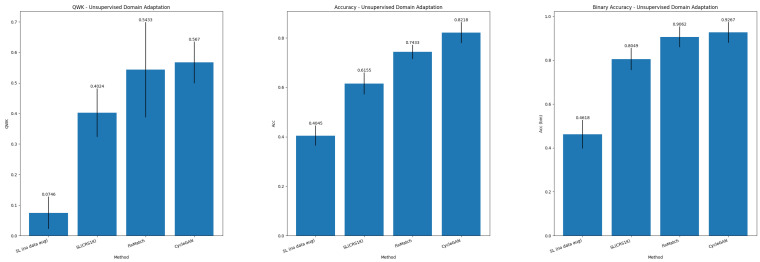
Summary of results for Unsupervised Domain Adaptation.

**Figure 5 sensors-25-02856-f005:**
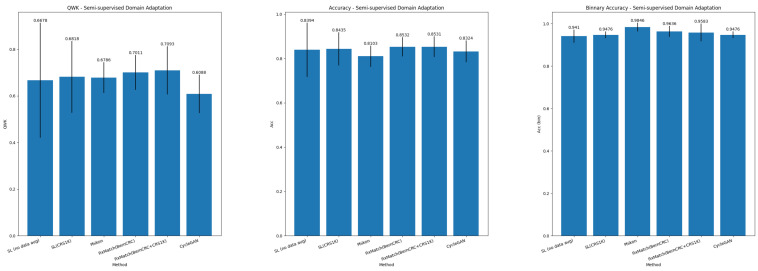
Summary of results for Semi-supervised Domain Adaptation.

**Table 1 sensors-25-02856-t001:** Data augmentation tailored to CPath as proposed in the work by Tellez et al. [31].

Transform	Confounding Factor(s)	Hyperparameters
Hue Shift	Staining	shift limits =[−0.125,0.125]
Contrast Shift	Staining/Whole-slide scanner	shift limits =[−0.2,0.2]
Saturation Shift	Staining/Whole-slide scanner	shift limits =[−0.125,0.125]
HED Jitter	Staining	α=0.1β=0.0075
Flips (Horizontal/Vertical)	Viewpoint	-
Rotation	Viewpoint	limits = $\left[-90^{\circ },90^{\circ }\right]$
Scaling	Whole-slide scanner	limits = [0.8,1.2]
Gaussian Blur	Whole-slide scanner	max kernel size = 15, σ limits = [0.1,2]
Gaussian Noise	Whole-slide scanner	μ=0,σ=[0,0.1]

**Table 2 sensors-25-02856-t002:** Results on the source domain (CRS10K) test set for grading of colorectal biopsies and polypectomies into Non-neoplastic (NNeo); Low-grade (LG), and High-grade (HG) lesions. We report the mean and standard deviation of the k-fold (k = 5) cross-validation on the CRS10K test set. Values in bold indicate the best result.

Method	MIL Train/Val Set	QWK	Acc	AUC (NNeo vs. Rest)	AUC (LG vs. Rest)	AUC (HG vs. Rest)
SL (no data aug)	CRS1K	0.8759	0.8919	0.9911	0.9627	0.9661
SL (CRS1K)	CRS1K	0.8977	0.9097	0.9955	0.9707	0.9780
FixMatch	CRS1K	0.9120 ± 0.0096	0.9222 ± 0.0052	**0.9956 ± 0.0011**	0.9768 ± 0.0021	0.9815 ± 0.0031
FixMatch	BernCRC+CRS1K	**0.9189 ± 0.0086**	**0.9289 ± 0.0060**	0.9953 ± 0.0006	**0.9781 ± 0.0016**	**0.9834 ± 0.0011**

**Table 3 sensors-25-02856-t003:** Results on the source domain (CRS10K) test set for binary classification of colorectal dysplasia. To perform binary classification, we combine the Low-grade (LG) and High-grade (HG) labels into a single class. We perform K-fold cross-validation and report the mean and standard deviation on the CRS10K test set. Values in bold indicate the best result.

Method	Train/Val set	Acc (Bin)	Sens	Spec	Prec	F1
SL (no data aug)	CRS1K	0.9688	0.9819	0.9157	0.9792	0.9464
SL (CRS1K)	CRS1K	0.9744	0.9847	0.9326	0.9833	0.9573
FixMatch	CRS1K	0.9792 ± 0.0044	0.9864 ± 0.0027	0.9506 ± 0.0157	0.9877 ± 0.0038	0.9688 ± 0.0100
FixMatch	BernCRC+CRS1K	**0.9810 ± 0.0028**	**0.9878 ± 0.0010**	**0.9539 ± 0.0120**	**0.9886 ± 0.0030**	**0.9709 ± 0.0076**

**Table 4 sensors-25-02856-t004:** Fréchet Inception Distance (FID) between BernCRC (target) transformed to CRS10K (source) and BernCRC (target) to CRS10K (source).

FID BernCRC (Target) Transformed—CRS10K (Source)	FID BernCRC (Target)—crs10k (Source)
9.9940	21.2732

**Table 5 sensors-25-02856-t005:** Results of Unsupervised Domain Adaptation (UDA) for colorectal tissue classification into Non-neoplastic (NNeo); Low-grade (LG); and High-grade (HG) lesions. We report the mean and standard deviation of the k-fold (k = 5) cross-validation on BernCRC. Values in bold indicate the best result.

Method	MIL Train/Val Set	QWK	Acc	AUC (NNeo vs. Rest)	AUC (LG vs. Rest)	AUC (HG vs. Rest)
SL (no data aug)	CRS1K	0.0746 ± 0.0529	0.4045 ± 0.0409	0.7034 ± 0.0800	0.5593 ± 0.0712	0.3055 ± 0.0433
SL (CRS1K)	CRS1K	0.4024 ± 0.0792	0.6155 ± 0.0443	0.9302 ± 0.0343	0.8025 ± 0.0793	0.7159 ± 0.1220
FixMatch	CRS1K	0.5433 ± 0.1561	0.7433 ± 0.0288	0.9607 ± 0.0288	0.8330 ± 0.0887	**0.8187 ± 0.1128**
CycleGAN	CRS1K	**0.5670 ± 0.0685**	**0.8218 ± 0.0424**	**0.9748 ± 0.0091**	**0.8654 ± 0.0668**	0.7435 ± 0.0814

**Table 6 sensors-25-02856-t006:** Results of the Unsupervised Domain Adaptation (UDA) for binary classification of colorectal dysplasia. To perform binary classification, we combine the Low-grade (LG) and High-grade (HG) labels into a single class. We report the mean and standard deviation of the k-fold (k = 5) cross-validation in BernCRC. Values in bold indicate the best result.

Method	MIL Train/Val Set	Acc (bin)	Sens	Spec	Prec	F1
SL (no data aug)	CRS1K	0.4618 ± 0.0649	0.4000 ± 0.0895	0.8333 ± 0.1491	0.9389 ± 0.0560	0.8797 ± 0.1084
SL (CRS1K)	CRS1K	0.8049 ± 0.0502	0.7835 ± 0.0630	0.9333 ± 0.0817	0.9856 ± 0.0177	0.9576 ± 0.0519
FixMatch	CRS1K	0.9062 ± 0.0472	**0.9265 ± 0.0558**	0.8000 ± 0.2667	0.9650 ± 0.04608	0.8529 ± 0.2030
CycleGAN	CRS1K	**0.9267 ± 0.0473**	0.91362 ± 0.0564	**1 ± 0**	**1 ± 0**	**1 ± 0**

**Table 7 sensors-25-02856-t007:** Results of Semi-supervised Domain Adaptation (SSDA) for grading of colorectal biopsies and polypectomies into Non-neoplastic (NNeo); Low-grade (LG); and High-grade (HG) lesions. We report the mean and standard deviation of the k-fold (k = 5) cross-validation in BernCRC. Values in bold indicate the best result.

Method	MIL Train/Val Set	QWK	Acc	AUC (NNeo vs. Rest)	AUC (LG vs. Rest)	AUC (HG vs. Rest)
SL (no data aug)	BernCRC	0.6678 ± 0.2469	0.8394 ± 0.1228	0.9277 ± 0.0434	0.8236 ± 0.1248	0.8450 ± 0.1641
SL (CRS1K)	BernCRC	0.6816 ± 0.1541	0.8435 ± 0.0747	0.98662 ± 0.0107	0.8238 ± 0.1005	0.7767 ± 0.1707
Phikon	BernCRC	0.6786 ± 0.0657	0.8103 ± 0.0476	**0.9949 ± 0.0078**	**0.8793 ± 0.0287**	**0.8283 ± 0.0395**
FixMatch	BernCRC	0.7011 ± 0.0748	**0.8532 ± 0.0431**	0.9906 ± 0.0086	0.8503 ± 0.0630	0.8181 ± 0.0645
FixMatch	BernCRC+CRS1K	**0.7093 ± 0.1037**	0.8531 ± 0.0464	0.9865 ± 0.0145	0.8466 ± 0.0484	0.8241 ± 0.0573
CycleGAN	BernCRC	0.6088 ± 0.0828	0.8324 ± 0.0488	0.9914 ± 0.0095	0.8642 ± 0.0754	0.8071 ± 0.1150

**Table 8 sensors-25-02856-t008:** Results of Semi-supervised Domain Adaptation (SSDA) for binary classification of colorectal dysplasia. To perform binary classification, we combine the Low-grade (LG) and High-grade (HG) labels into a single class. We report the mean and standard deviation of the k-fold (k = 5) cross-validation in BernCRC. Values in bold indicate the best result.

Method	MIL Train/Val Set	Acc (Bin)	Sens	Spec	Prec	F1
SL (no data aug)	BernCRC	0.9410 ± 0.0312	0.9812 ± 0.0375	0.7000 ± 0.1265	0.9523 ± 0.0218	0.8016 ± 0.0954
SL (CRS1K)	BernCRC	0.9476 ± 0.0158	0.9691 ± 0.0198	0.8201 ± 0.1066	0.9699 ± 0.0186	0.8860 ± 0.0699
Phikon	BernCRC	**0.9846 ± 0.0205**	**0.9939 ± 0.0121**	**0.9333 ± 0.0817**	**0.9881 ± 0.0146**	**0.9586 ± 0.0507**
FixMatch	BernCRC	0.9636 ± 0.0262	0.9814 ± 0.0249	0.8600 ± 0.1272	0.9763 ± 0.0221	0.9108 ± 0.0828
FixMatch	BernCRC+CRS1K	0.9583 ± 0.0419	0.9691 ± 0.0275	0.9000 ± 0.1333	0.9814 ± 0.0245	0.9351 ± 0.0875
CycleGAN	BernCRC	0.9476 ± 0.0158	0.9629 ± 0.0230	0.8600 ± 0.1272	0.9761 ± 0.0217	0.9107 ± 0.0826

## Data Availability

The original CRS10K data presented in the study are openly available in INESC TEC research data repository at https://doi.org/10.25747/fb1q-j507 (accessed on 28 April 2025). The BernCRC dataset presented in this article is not readily available because of the informed user consent and the data transfer agreement that allow us to use the data for research, but we do not have permission to make the data publicly available.

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
