# Peer review of "Bridging Domain Gaps in Computational Pathology: A Comparative Study of Adaptation Strategies"

_sensors, 2025, doi:10.3390/s25092856_

Round 1

Reviewer 1 Report

Comments and Suggestions for Authors
  1. The introduction provides a comprehensive explanation of the importance of domain adaptation in computational pathology and summarizes the limitations of existing methods. However, the description of specific cross-domain challenges remains relatively general, lacking an in-depth discussion of issues such as staining variability, image blurriness, and resolution inconsistencies. These factors significantly affect the generalization ability of deep learning models. It is recommended that the authors supplement this section by highlighting the limitations of mainstream domain adaptation methods in addressing these challenges and exploring potential improvements.

  1. The experiments adopt the CRS10K and BernCRC datasets, which are representative and challenging. However, the current description of data preprocessing and augmentation strategies is mainly descriptive, lacking quantitative analysis of their impact on model performance. It is recommended to add an experimental analysis comparing the effects of different preprocessing and augmentation combinations on feature extraction and downstream classification tasks to demonstrate the effectiveness of the current preprocessing pipeline.

  1. The paper uses FixMatch, CycleGAN, and Phikon as the main comparison methods, which is innovative to some extent. However, the methods section lacks in-depth theoretical explanations and practical applicability analysis, especially regarding how staining variability and tissue structural complexity, which are unique to computational pathology, impact the adaptation performance of each method. It is recommended to include a more detailed analysis of the adaptation mechanisms of each method in specific pathology imaging scenarios and to further discuss the potential complementarity of these methods.

  1. Computational Pathology needs to draw on the latest research trends in machine learning and should at least be mentioned in the analysis, multitasking and multimodal machine learning algorithms. the author can analyze multi task learning for time estimation and identity recognition of hand heat traces, deep soft threshold feature separation networks for infrared fingerprint identity recognition and time estimation, and related research ideas, providing inspiration for the ideas of this study. Meanwhile, the author should more effectively describe the practical application scope and significance of this article.

  1. The conclusion summarizes the performance of FixMatch and CycleGAN across different tasks, but the outlook for future work is relatively broad. Considering the main challenges in domain adaptation for computational pathology, such as hidden stratification, batch effects, and stain normalization, it is recommended to propose more targeted and practical future research directions based on the findings of this study to enhance the practical guidance of the research.
Comments on the Quality of English Language

The English could be improved to more clearly express the research.

Reviewer 2 Report

Comments and Suggestions for Authors

Thanks for the manuscript. Here are some points to improve it:

  1. It is suggested that the authors complement the QWK metric with the Frechet Inception Distance (FID) and Kernel Inception Distance (KID) metrics, which would explicitly assess the visual quality and realism of the domain transfer performed by CycleGAN.
  2. Improve your presentation results. Add charts and visual summaries that complement your long tables or instead organize long tables systematically for clarity. Make the results more straightforward in the main text to improve the readability.
  3. Could you provide a detailed justification or explain the reasoning concerning hyperparameter selection?
  4. You can improve your Discussion section by comparing your results against the literature.
  5. Improve your Introduction section and abstract, detailing the study gaps/questions you pretend to research and address. Now, it is a little difficult to understand it. 

Round 2

Reviewer 1 Report

Comments and Suggestions for Authors The revised version has a very good improvement in algorithm and logic. I warmly recommend publication in present form.

Author Response

Comments: The revised version has a very good improvement in algorithm and logic. I warmly recommend publication in present form.

Response: We thank the reviewer for the very positive feedback. We appreciate the time and effort that the reviewer dedicated to providing feedback on our manuscript and are grateful for the insightful comments on and valuable improvements to our paper. 

Reviewer 2 Report

Comments and Suggestions for Authors

The authors addressed all comments. However, improving the discussion section or mentioning the limitations you expressed in the cover letter would be desirable. Thanks. 

Author Response

Thank you for giving us the opportunity to submit a revised draft of the  manuscript “Bridging Domain Gaps in Computational Pathology: A Comparative Study of Adaptation Strategies.”.  We appreciate the time and effort that the reviewers dedicated to providing feedback on our manuscript and are grateful for the insightful comments on and valuable improvements to our paper. We have incorporated most of the suggestions made by the reviewers. Those changes are highlighted within the manuscript. Please see below in blue for a point-by-point response to the reviewers’ comments and concerns. All page numbers refer to the revised manuscript file with tracked changes.

"The authors addressed all comments. However, improving the discussion section or mentioning the limitations you expressed in the cover letter would be desirable. Thanks. "

Comment: We thank the reviewer for the constructive comment.

Action: We have improved the discussion section as follows: (page 13, lines 323 -329) “Another finding of this work is that data augmentation significantly improves performance compared to no data augmentation, with a 32.78% increase in QWK and a 21.1% increase in accuracy. However, a limitation of this study is that we only tested one combination of hyperparameters for data augmentation due to the long run times, which may lead to suboptimal results. Furthermore, it would be beneficial to compare our results with the existing literature. Unfortunately, there is limited research on domain adaptation in computational pathology, which makes direct comparisons difficult.”